# Intractable Itch in Atopic Dermatitis: Causes and Treatments

**DOI:** 10.3390/biomedicines9030229

**Published:** 2021-02-25

**Authors:** Yoshie Umehara, Chanisa Kiatsurayanon, Juan Valentin Trujillo-Paez, Panjit Chieosilapatham, Ge Peng, Hainan Yue, Hai Le Thanh Nguyen, Pu Song, Ko Okumura, Hideoki Ogawa, François Niyonsaba

**Affiliations:** 1Atopy (Allergy) Research Center, Juntendo University Graduate School of Medicine, Tokyo 113-8421, Japan; y-umeha@juntendo.ac.jp (Y.U.); taneiro87@hotmail.com (J.V.T.-P.); g-peng@juntendo.ac.jp (G.P.); yhn125300@163.com (H.Y.); haidalieuhue0710@gmail.com (H.L.T.N.); kokumura@juntendo.ac.jp (K.O.); ogawa@juntendo.ac.jp (H.O.); 2Institute of Dermatology, Department of Medical Services, Ministry of Public Health, Bangkok 10400, Thailand; chanisa.kiatsurayanon@gmail.com; 3Department of Microbiology, Faculty of Medicine, Chiang Mai University, Chiang Mai 50200, Thailand; chillipop4507@gmail.com; 4Department of Dermatology, Xijing Hospital, Fourth Military Medical University, Xi’an 710032, China; songpu@fmmu.edu.cn; 5Faculty of International Liberal Arts, Juntendo University, Tokyo 113-8421, Japan

**Keywords:** atopic dermatitis, dry skin, intractable itch, itch/pruritus, keratinocyte, sensory nerve fiber

## Abstract

Itch or pruritus is the hallmark of atopic dermatitis and is defined as an unpleasant sensation that evokes the desire to scratch. It is also believed that itch is a signal of danger from various environmental factors or physiological abnormalities. Because histamine is a well-known substance inducing itch, H_1_-antihistamines are the most frequently used drugs to treat pruritus. However, H_1_-antihistamines are not fully effective against intractable itch in patients with atopic dermatitis. Given that intractable itch is a clinical problem that markedly decreases quality of life, its treatment in atopic dermatitis is of high importance. Histamine-independent itch may be elicited by various pruritogens, including proteases, cytokines, neuropeptides, lipids, and opioids, and their cognate receptors, such as protease-activated receptors, cytokine receptors, Mas-related G protein-coupled receptors, opioid receptors, and transient receptor potential channels. In addition, cutaneous hyperinnervation is partly involved in itch sensitization in the periphery. It is believed that dry skin is a key feature of intractable itch in atopic dermatitis. Treatment of the underlying conditions that cause itch is necessary to improve the quality of life of patients with atopic dermatitis. This review describes current insights into the pathophysiology of itch and its treatment in atopic dermatitis.

## 1. Introduction

Patients with atopic dermatitis (AD) suffer from recurrent dermatitis and intractable itch. The most frequent clinical phenotype of AD is lichenified/exudative flexural dermatitis alone or associated with head/neck eczema or hand eczema [1]. Although there is no mortality directly associated with AD, this condition substantially impacts patients’ quality of life. In addition to the social stigmatization due to visible skin lesions, severe pruritus can disrupt sleep in patients with AD, which can lead to psychosocial comorbidities, including depression, anxiety, and suicidal ideations [2]. The economic burden of AD is attributed to not only direct medical costs but also to costs through lost work productivity. For instance, in Europe, the direct annual costs have been estimated to EUR 7000, while the indirect costs ranged from EUR 7000 to EUR 14,000 per patient with severe AD [2]. Taken together, patients with AD and society will benefit from the reduction of the burden caused by economic and psychosocial comorbidities in AD.

Although the pathogenesis of AD has not been fully elucidated, it is influenced in a complicated manner by genetic factors, immune dysfunction, physical conditions, stress and weather [3,4,5]. Furthermore, the observation that AD prevalence appears to be increasing in developing countries suggests an important role of environmental factors in the pathogenic mechanism of AD [6]. Because histamine is a well-known pruritogen that is involved in itch accompanied by urticaria caused by the degranulation of mast cells, H_1_-antihistamines are the most frequently used drugs to treat pruritus. However, H_1_-antihistamines, even at high doses, are not fully effective against intractable itch in patients with AD [7]. Since severe pruritus leads to disturbances in patient sleep and work that markedly decrease quality of life, H_1_-antihistamine-resistant itch is a clinical problem in AD. Thus, understanding the pathogenesis of itch and its treatment is important.

This review highlights recent knowledge regarding the mechanisms of itch in AD and itch treatment.

## 2. Methods

For the current review, relevant literature published in English was searched for using PubMed, Google Scholar and Web of Science to identify systematic articles published prior to January 2021. The terms used for retrieval were “dry skin”, “intractable itch”, “itch”, “pruritus”, “keratinocyte”, “sensory nerve fiber” etc. combined with “atopic dermatitis” as keywords. We screened the articles to select those published in international peer-reviewed journals or books. Discrepancies in the assessment were resolved through discussion among the authors.

## 3. Transduction of Itch

It is believed that itch is a signal of danger from various environmental factors or physiological abnormalities. Itch initially originates from pruritogens, which are caused by inflammation, dryness or other skin damage. Pruritogens activate certain receptors on the free nerve endings of sensory neurons. Itch sensation is mediated by primary (peripheral) sensory afferents, especially C-fibers, which have cell bodies in the dorsal root ganglia or trigeminal ganglia [8,9]. Pruritogen receptors are mostly G protein-coupled receptors (GPCRs) that promote the opening of the ion channel group transient receptor potential cation channels, especially transient receptor potential vanilloid 1 (TRPV1) and transient receptor potential ankyrin 1 (TRPA1) [10]. TRPV1 and TRPA1 are activated by diverse stimuli in addition to GPCR ligands, including extracellular pH, adenosine triphosphate (ATP), prostaglandins, oxidants, capsaicin, allyl isothiocyanate, heat and cold, and TRPV1 and TRPA1, which are involved in the pathology of AD and pruritus [11,12]. The pruritus signal of peripheral sensory afferents is transmitted through the spinal cord in the dorsal root ganglia or trigeminal ganglia to the somatosensory cortex, resulting in recognition of an itch sensation.

In atopic skin, increased penetration of pathogens and antigens and nerve fiber density with pruritogen receptors elicit severe pruritus following the skin barrier dysfunction or inflammation (Figure 1). In addition, the reduced itch threshold of atopic skin compared with that of healthy skin provokes an abnormal itch sensation called “alloknesis” and “hyperknesis”. Alloknesis is a pathology in which pruritus is elicited by innocuous mechanical stimulation, including contact with fabrics, dressing and undressing [13]. In contrast, hyperknesis represents increased itch elicited by pruritogens, and the sensation of pruritus is stronger in patients with AD than in healthy individuals. Moreover, noxious stimuli evoke pruritus instead of pain in AD patients [14]. Therefore, in AD, itch sensation evokes the desire to scratch, and scratching induces the production of inflammatory cytokines by keratinocytes, further exacerbating eczema and inflammation in the skin (itch–scratch cycle) (Figure 2) [5,13].

In healthy skin, the penetration of pathogens or antigens and excessive water evaporation are prevented by the stratum corneum barrier and tight junction barrier. Furthermore, the cutaneous nerve fibers terminate in the dermis. In atopic skin, the transepidermal water loss (water evaporation) is increased and the damaged skin barrier enhances penetration of pathogens and antigens. Furthermore, recurrent dermatitis is accompanied by intractable itch, which is induced by inflammation and epidermal hyperinnervation.

Itch sensation evokes the desire to scratch, and scratching impairs skin barrier function. The barrier dysfunction leads to the production of inflammatory and itch mediators (cytokines, proteases, neuropeptides, lipids) by cutaneous cells, further exacerbating eczema and inflammation in the skin. Both barrier dysfunction and inflammation contribute to the increased number of fibers.

## 4. Substances and Receptors Inducing Itch in AD

### 4.1. Histamine

Histamine is involved primarily in acute itch and is a cause of itch in urticaria and insect bite reactions. Histamine is produced mainly by mast cells, although other cells, including basophils, keratinocytes and neurons in the skin, can also release it [15]. Although four receptors have been identified as histamine receptors, H1, H2, H3 and H4, only the H1 and H4 receptors activate TRPV1 and are considered therapeutic targets for pruritus [16]. However, H_1_-antihistamines do not improve intractable itch in patients with AD, implying that chronic itch is largely induced by a histamine-independent pathway [7,8]. The application of antihistamines to AD model mice is also known to be ineffective. In NC/Nga mice (which spontaneously develop AD-like skin lesions when they are raised in conventional circumstances), long-duration scratching (longer than one second) is not suppressed by the H_1_ receptor antagonist chlorpheniramine or the H_2_ receptor antagonist famotidine and the H_3/4_ receptor antagonist thioperamide [17]. Additionally, treatment with the H_4_ receptor antagonist JNJ7777120 or JNJ28307474 does not affect the inhibition of scratching behavior or amelioration of AD in NC/Nga mice [18].

### 4.2. Proteases

Several proteases produced by cutaneous cells or exogenous biotic factors, including bacteria, mites and plants, are involved in pruritus in AD. Endogenous proteases include tryptases, chymases, trypsins, and kallikreins and are produced by keratinocytes, mast cells, macrophages, dendritic cells, B cells, T cells, and neutrophils [19]. The number of tryptase-positive mast cells is increased in the upper dermis of the lesional and nonlesional skin of patients with AD [20]. In addition, various kallikrein proteases are upregulated in the stratum corneum of patients with AD [21].

Proteases cleave the extracellular domain of protease-activated receptors (PARs), and then the new amino terminus of the receptor itself acts as a PAR ligand. To date, four PARs, PAR-1, PAR-2, PAR-3, and PAR-4, have been identified. PAR-1, PAR-2 and PAR-4 are involved in acute itch in mice activated by the specific peptides TFLLR, SLIGRL, and AYPGKF, respectively [22]. PAR-1, PAR-2, and PAR-4 are expressed in cutaneous nerve fibers, keratinocytes, mast cells, and macrophages [19]. The number of PAR-2-positive keratinocytes and the activity of serine proteases are increased in the skin of AD model NC mice [23]. Plant cysteine proteases, including mucunain (cowhage), bromelain (pineapple stem), ficin (fig tree latex), and papain (papaya), induce itch via activation of PAR-2 and PAR-4 [24,25].

H_1_-antihistamines inhibit itch in mice elicited by agonists of PAR-1 and PAR-4 but not agonists of PAR-2, suggesting the involvement of PAR-2 in histamine-independent itch [22]. Given that scratching evoked by trypsin, which acts on PAR-1, PAR-2, and PAR-4, is suppressed by TRPV1 inhibition or TRPV1 knockout in mice, as well as by the H_1_-antihistamine cyproheptadine, trypsin-evoked pruritus is considered to be involved in histamine-dependent itch [26]. In PAR-2 knockout mice, trypsin- and SLIGRL-evoked scratching still occur; however, tryptase-evoked scratching is suppressed in these mice and is also inhibited by an anti-PAR-2 neutralizing antibody and the PAR-2 antagonist FSLLRY [27]. These findings suggest that different signaling pathways are activated by different agonists and/or that the roles of PAR-2 in neurons and keratinocytes differ. The serine protease inhibitor nafamostat mesilate inhibits tryptase-evoked scratching and spontaneous scratching in AD model NC mice [23,28]. An anti-PAR-2 antibody was also shown to suppress spontaneous scratching in NC mice [23]. Topical application of tacrolimus (the calcineurin inhibitor FK506) inhibits SLIGRL-evoked scratching, suggesting that its antipruritic effect is due to PAR-2 signaling inhibition [29].

### 4.3. Cytokines

Some cytokines are responsible not only for inflammation but also for itch sensation. Interleukin (IL)-31 is a cytokine mainly produced by type 2 helper T cells (Th2 cells) that exhibits upregulated expression in the lesional skin of patients with skin disease accompanied by itch, including AD, contact dermatitis and prurigo nodularis [30,31,32]. In addition to Th2 cells, mast cells, macrophages, dendritic cells, and eosinophils also produce IL-31 [33]. It has been reported that IL-31 participates in the itch sensation and promotes long-lasting scratching behavior in NC/Nga and BALB/c mice [34]. The IL-31 receptor is a heterodimer that consists of IL-31 receptor A and an oncostatin M receptor. IL-31 evokes itch directly via IL-31 receptor A expressed in sensory neurons via both TRPV1 and TRPA1 activation [35]. A missense mutation in the oncostatin M-specific receptor subunit β gene was found in families affected by familial primary localized cutaneous amyloidosis, an autosomal-dominant skin disease associated with chronic, severe pruritus [36]. It has been reported that the humanized anti-human IL-31 receptor A monoclonal antibody CIM331 (nemolizumab) reduces the severity and pruritus score of AD [37,38].

A second cytokine contributing to itch is thymic stromal lymphopoietin (TSLP), which is secreted primarily by keratinocytes, mast cells, and dendritic cells and promotes Th2 immune responses [39,40]. The expression of TSLP is upregulated in the epidermal keratinocytes of AD patients [41,42]. TSLP binds a heterodimeric receptor comprising the IL-7 receptor α chain and TSLP receptor expressed in dendritic cells, T cells, B cells, mast cells, basophils, and eosinophils [43,44]. TSLP directly elicits itch via the TSLP receptor on TRPA1-positive sensory neurons [42]. Interestingly, in return, scratching induces the production of TSLP by keratinocytes, leading to an itch–scratch cycle [45].

Both IL-4 and IL-13 are well-known Th2 cytokines overexpressed in AD skin. It has been shown that IL-4 and IL-13 are involved in chronic but not acute itch by directly activating sensory neurons via the IL-4 receptor α chain [46]. The IL-4 receptor α (IL-4Ra) chain is shared by IL-4 and IL-13 receptors and activates janus kinase (JAK) 1, suggesting that blockade of IL-4Ra or JAK1 signaling in sensory neurons might be a useful treatment for chronic itch. In fact, the fully human monoclonal antibody dupilumab, which blocks binding to IL-4Ra, reduces the severity and pruritus score of AD [47,48]. Moreover, baricitinib, a selective JAK1 and JAK2 inhibitor, reduces pruritus and inflammation in patients with moderate to severe AD [49]. In addition, IL-13 antibodies, lebrikizumab and tralokinumab, improve AD symptoms although they do not show strong effect on pruritus [50,51]. Overall, more biologic drugs blocking IL-4 and IL-13 are under development and will be available in the future for AD treatment [52,53,54].

### 4.4. Neuropeptides

Neuropeptides play key roles in modulating neuronal activity. Substance P is a tachykinin neuropeptide that is involved in AD itch [55]. Substance P is secreted by sensory neurons and keratinocytes and elicits itch by histamine-dependent and histamine-independent mechanisms. A relatively high density of substance-P-containing nerve fibers has been observed in AD skin [56]. Neurokinin 1 receptor (NK1R) is a substance P receptor involved in itch and is expressed in sensory neurons, mast cells, keratinocytes, and fibroblasts. These cells release additional mediators inducing pruritus following stimulation with substance P. Although some patients did not respond, a study showed that the oral neurokinin 1 receptor antagonist aprepitant demonstrated efficacy in the treatment of intractable pruritus in AD and prurigo nodularis [57].

Substance P and calcitonin-gene-related peptide (CGRP) have roles in itch sensation hypersensitivity and neurogenic inflammation [58]. The release of both substance P and CGRP from sensory neurons excited by histamine leads to local vasodilation, plasma extravasation, and mast cell degranulation, whose response is neurogenic inflammation [59,60]. These findings indicate that neuropeptides play important roles in chronic itch under the pathological conditions of AD.

### 4.5. Lipids

Lipid mediators are produced by various pathophysiological stimuli that contribute to the pathogenesis of various inflammatory skin diseases. The roles of lipid mediators in the pathogenesis of AD and intractable pruritus remain largely unknown. Arachidonic acid released from membrane phospholipids is converted by each synthase into lipid mediators, including prostanoids, leukotrienes (LTs), and hydroxy-eicosatetraenoic acids [61]. Prostanoids, encompassing prostaglandin (PG) and thromboxane (TX), are synthesized by cyclooxygenase. Finally, arachidonic acid is metabolized to PGD_2_, PGE_2_, PGF_2α_, PGI_2_, and TXA_2_ by their specific synthesis pathways. Prostanoids are released from cells immediately after synthesis and are chemically and metabolically unstable. These lipids thereby act on target cells only locally via GPCRs [61]. The concentrations of PGE_2_ and LTB_4_ are elevated in the lesional skin of patients with AD, suggesting that these mediators are involved in biochemical processes leading to AD through cutaneous inflammation [62]. It has been demonstrated that PGE_2_ acts as a weak pruritogen and potent vasodilator in normal skin as well as in the skin of patients with AD without induction of protein extravasation [63]. Furthermore, an antagonist of LTB_4_ receptor, ONO-4057, has been shown to inhibit spontaneous scratching of NC mice with chronic dermatitis [64]. A stable analog of TXA_2_, U-46619, has been shown to elicit itch in mice through thromboxane prostanoid receptors expressed in cutaneous nerve fibers and keratinocytes [65]. This TXA_2_-induced scratching behavior is inhibited by the thromboxane prostanoid receptor antagonist ONO-3708 and thromboxane prostanoid receptor deficiency in mice [65].

On the other hand, topical application of PGD_2_, PGI_1_, PGE_1_, PGE_2_ or arachidonic acid was shown to suppress scratching in NC/Nga mice with AD, while the cyclooxygenase inhibitor indomethacin enhanced scratching [66]. Among the above lipid mediators, PGD_2_ is the most effective mediator, and its inhibition depends on the prostanoid DP_1_ receptor but not on the DP_2_ receptor, indicating the therapeutic potential of prostanoid DP_1_ receptor agonists [66].

### 4.6. Opioids

Opioids are peptides that have pharmacological effects similar to those of morphine. The endogenous opioids β-endorphin and dynorphins activate the µ-opioid receptor and κ-opioid receptor, and the balance of this activation plays pivotal roles in the regulation of pruritus in both the central and peripheral nervous systems [67]. Opioid receptors are GPCRs that also recognize exogenous opioids, such as opiates and alkaloids. It is thought that opioid receptors can contribute to cell differentiation, migration, wound healing, and immunity in human skin [68]. The expression of µ-opioid receptor has been observed in the epidermal keratinocytes and primary sensory afferents of normal skin [69,70]. The serum concentration of β-endorphin is increased in patients with AD and is correlated with both itch intensity and disease severity [71]. Opioid-induced itch is a well-known side effect of pain treatment with morphine, a µ-opioid receptor agonist. In contrast, the µ-opioid receptor antagonist naloxone and κ-opioid receptor agonist nalfurafine decrease pruritus in patients with AD, chronic renal failure or cholestasis [68]. Furthermore, topical application of the µ-opioid receptor antagonist naltrexone also inhibits pruritus in patients with AD [72]. Phototherapy, such as ultraviolet A (UVA) treatment, UVA1 treatment, narrow band (NB)-UVB treatment and excimer lasers or lamps, improves pruritus and dermatitis [73,74]. It has been reported that dynorphin levels are downregulated in the epidermis of AD patients and that psoralen-ultraviolet A (PUVA) therapy rescues the downregulation and visual analog scale (VAS) scores [75]. Thus, some studies suggest that opioids may be directly associated with modulation of itch.

## 5. Cutaneous Nerve Fibers

Various receptors related to itch are expressed on cutaneous nerve fibers that terminate in the dermis of healthy skin. An increased epidermal nerve fiber density is observed in patients with AD and animal models of AD, suggesting that a relatively high nerve fiber density is partly responsible for pruritus in the skin [76,77]. The increased numbers of nerve fibers in the epidermis are susceptible to eliciting itch in response to exogenous mechanical, chemical or biological stimuli or endogenous pruritogens [78]. It is believed that the elongation and high density of cutaneous nerve fibers are caused by an imbalance among axonal guidance molecules, nerve elongation factors and nerve repulsion factors produced by epidermal keratinocytes [79]. Nerve growth factor (NGF) is a major nerve elongation factor involved in nerve growth and repair, and its expression is increased in keratinocytes, mast cells and the serum in AD [80]. It has been shown that the plasma levels of substance P and NGF are useful markers of disease activity in AD [80,81]. The local concentration of NGF is higher in the lesional skin of AD than in healthy skin. Artemin is another nerve elongation factor that accumulates in the lesional skin of patients with AD, suggesting its involvement in the pathogenesis of AD. Furthermore, artemin is involved in epidermal hyperinnervation and hypersensitivity to warm sensations, mimicking the warmth-induced itch observed in AD [82,83].

Semaphorin 3A is a nerve repulsion factor that is mainly distributed in the basal layer of normal skin and inhibits aberrant penetration of nerve fibers into the epidermis [84]. However, the expression of semaphorin 3A is decreased in the skin of AD patients compared with that of healthy controls. Notably, the cutaneous nerve density is modulated by the balance between nerve elongation factor expression and nerve repulsion factor expression. Similar findings have been reported in the lesional skin of AD model NC/Nga mice [77]. Therefore, regression of the increased density of nerve fibers is expected to be effective against itch, and normalization of imbalances in the expression levels of nerve elongation factors and nerve repulsion factors may be a useful treatment approach.

Cyclosporine A is an immunosuppressant used for the treatment of inflammatory diseases, preventing rejection of allogeneic transplants and the treatment of severe AD. Intraperitoneal injection of cyclosporine A was shown to reduce the epidermal nerve fiber density, scratching numbers and dermatitis scores in NC/Nga mice with AD [85]. Phototherapy, specifically PUVA and NB-UVB irradiation, also cause diminution of the epidermal nerve density and pruritus by normalizing imbalances in the expression levels of axon guidance molecules in AD lesional skin [86,87]. Moreover, excimer lamp irradiation has antipruritic effects induced by causing epidermal nerve fiber degeneration [88].

## 6. Skin Dryness

Dryness is a common finding in the skin of patients with AD due to defects in barrier function [89]. Dysfunction of the skin barrier leads to a loss of essential internal water in the skin, resulting in increased transepidermal water loss (TEWL) and decreased stratum corneum hydration. In addition, skin dryness causes various physiological responses. Elevated TEWL and increased TSLP expression in human skin with the barrier function disrupted by tape stripping have been reported [90]. Furthermore, it has been shown that hypertrophy and degranulation of dermal mast cells caused by exposure to low humidity correlate with seasonal exacerbation of itch and dermatitis [91]. Because the granules of mast cells contain histamine, serotonin, LTB_4_, proteases, and pruritogens, xerosis is associated with diverse induction processes of pruritus.

Repeated application of the surfactant sodium dodecyl sulfate or acetone, diethyl ether and water (AEW) to mouse skin induces dry skin characterized by increased TEWL and scratching behavior [92,93]. In AEW-treated dry skin mice, scratching behavior is inhibited by the opioid antagonists naloxone and naltrexone, but these drugs are not effective in mast-cell-deficient mice [92]. Moreover, spontaneous scratching can be attenuated with an anti-PAR-2 antibody in AEW-induced mice [94]. This model mouse exhibits alloknesis and hyperknesis, suggesting that the abnormal itch sensations in AD are partly caused by modulating skin dryness [94,95,96]. These studies indicate the release of various endogenous mediators even under only dry skin conditions.

## 7. Dysregulation of the Expression of Antimicrobial Peptides

Antimicrobial peptides or host defense peptides have the ability to kill or inactivate pathogenic microorganisms. In the skin, these molecules are mainly produced by keratinocytes as part of innate immune system [97]. A large number of antimicrobial peptides have been identified in the human skin, and among them the human β-defensins (hBDs), cathelicidin LL-37, and S100 proteins are well-characterized. These peptides can be either constitutively expressed or inducible following infection, tissue damage, and pro-inflammatory cytokines [98]. In addition to their broad-spectrum antimicrobial activity, the skin-derived antimicrobial peptides control diverse biological functions, such as inflammatory response regulation, cytokine and chemokine production, cell migration, proliferation, and promotion of angiogenesis and wound healing [98,99].

The induction of antimicrobial peptides in the epidermal keratinocytes is impaired in lesional skin in patients with AD compared to patients with psoriasis (which is associated with a comparable degree of epidermal barrier impairment as in AD), explaining the frequent bacterial and viral infections, particularly with *Staphylococcus aureus*, in patients with AD [100,101]. The reduced expression of antimicrobial peptides in AD is partly due to the increased production of Th2 cytokines such as IL-4 and IL-13 that inhibit the expression of antimicrobial peptides [102,103]. In contrast to AD, high levels of antimicrobial peptides and the low prevalence of infections have been observed in psoriatic lesions [100,102]. The expression of hBDs, LL-37 and S100A, in psoriatic skin is enhanced by IL-1β, IL-17A, IL-22, and interferon-γ [98,104]. Therefore, restoration of antimicrobial peptides in the skin of patients with AD might play a protective role against infections in these patients.

In addition to protecting against infections, antimicrobial peptides are thought to restore the damaged skin barrier in AD. In fact, hBD-3, LL-37, and S100A7 have been found to improve the skin barrier function through induction of distribution/localization of tight junction proteins in human epidermal keratinocytes [105,106,107]. Furthermore, antimicrobial peptides control the skin innervation by regulating the expression of axon guidance molecules. For example, LL-37 has been shown to upregulate the nerve repulsion factor semaphorin 3A in keratinocytes [108], while both hBD-3 and LL-37 suppress the production of the nerve elongation factors artemin and NGF [109]. These findings suggest that antimicrobial peptides might be potential therapeutic agents for AD. However, antimicrobial peptides may also be harmful in AD. For instance, it has been found that hBDs and LL-37 promote the release of inflammatory mediators such as histamine, eicosanoids, and IL-31 from mast cells and increase vascular permeability [110,111,112]. Moreover, hBDs stimulate T cells to produce IL-4, IL-13, and IL-31, which are implicated in the pathogenesis of AD [113]. Furthermore, the expression levels of hBD-2 are higher in the lesional skin compared with nonlesional skin of AD and correlate with the disease severity and TEWL [114]. Taken together, these studies suggest that antimicrobial peptides could potentially be a double-edged sword in the pathogenesis of AD through regulation of skin inflammation, barrier function, and innervation.

## 8. Conclusions

Intractable and intense itch is a remarkable symptom of AD that markedly affects patients’ quality of life. Given that itch evokes scratching and worsens inflammatory eczema in AD skin, its alleviation is important in the treatment of AD. The mechanisms of pathogenesis and exacerbation of AD are still unclear. Although the number of studies related to the mechanisms of itch and transduction is increasing rapidly, therapeutic agents for intractable itch still need to be developed. The development of novel therapeutic strategies for patients with AD should involve management of pruritus, leading to an improvement in patient quality of life.

## Figures and Tables

**Figure 1 biomedicines-09-00229-f001:**
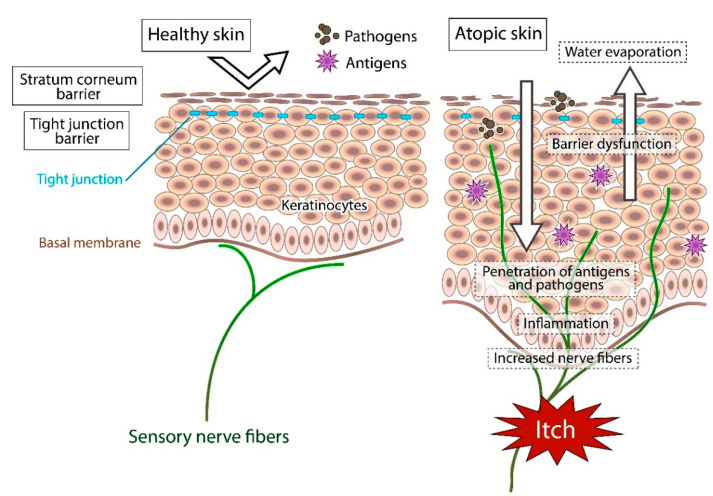
Schematic representation of the pathogenesis of itch in atopic dermatitis.

**Figure 2 biomedicines-09-00229-f002:**
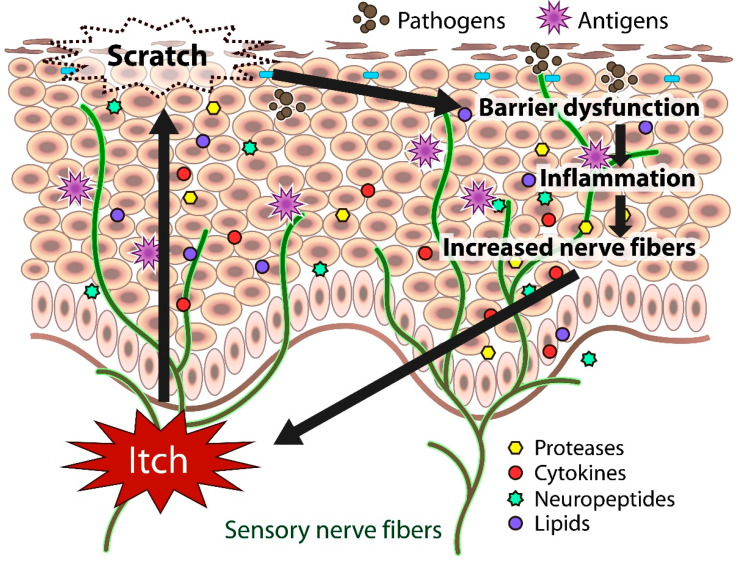
Schematic representation of chronic itch in atopic dermatitis.

## Data Availability

Data sharing not applicable.

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
