# Peer review of "Intractable Itch in Atopic Dermatitis: Causes and Treatments"

_biomedicines, 2021, doi:10.3390/biomedicines9030229_

Round 1
Reviewer 1 Report
The work is interesting and well written. there aren't many corrections to make, mostly suggestions.
Emphasize the burden of itching in atopic dermatitis and the economic impact it has on the lives of patients and society. The work Girolomoni G, Luger T, Nosbaum A, Gruben D, Romero W, Llamado LJ, DiBonaventura M. The Economic and Psychosocial Comorbidity Burden Among Adults with Moderate-to-Severe Atopic Dermatitis in Europe: Analysis of a Cross- Sectional Survey. Dermatol Ther (Heidelb). 2020 Nov 12. doi: 10.1007 / s13555-020-00459-8. Epub ahead of print. PMID: 33180320.
In the section where the authors talk about possible tereputic targets, I suggest to the authors to consult and cite two works: "Fabbrocini G, Napolitano M, Megna M, Balato N, Patruno C. Treatment of Atopic Dermatitis with Biologic Drugs. Dermatol Ther (Heidelb ). 2018 Dec; 8 (4): 527-538. Doi: 10.1007 / s13555-018-0258-x. Epub 2018 Sep 4. PMID: 30182182; PMCID: PMC6261117. ", And" Napolitano M, Marasca C, Fabbrocini G, Patruno C. Adult atopic dermatitis: new and emerging therapies. Expert Rev Clin Pharmacol. 2018 Sep; 11 (9): 867-878. Doi: 10.1080 / 17512433.2018.1507734. Epub 2018 Aug 9. PMID: 30073901. "
Author Response
Reviewer 1
Comments and Suggestions for Authors
The work is interesting and well written. there aren't many corrections to make, mostly suggestions.
Answer
We would like to thank the Reviewer 1 for taking his/her time to read our manuscript, and for kind appreciation on this work. Below, we provide point-by-point responses to your comments.
Comment #1: Emphasize the burden of itching in atopic dermatitis and the economic impact it has on the lives of patients and society. The work Girolomoni G, Luger T, Nosbaum A, Gruben D, Romero W, Llamado LJ, DiBonaventura M. The Economic and Psychosocial Comorbidity Burden Among Adults with Moderate-to-Severe Atopic Dermatitis in Europe: Analysis of a Cross- Sectional Survey. Dermatol Ther (Heidelb). 2020 Nov 12. doi: 10.1007 / s13555-020-00459-8. Epub ahead of print. PMID: 33180320.
Answer to comment #1
We fully agree with the Reviewer’s suggestion that the socio-economic impact of atopic dermatitis should have been added in this manuscript. As seen on line 36-44, we cited that “The most frequent clinical phenotype of AD is lichenified/exudative flexural dermatitis alone or associated with head/neck eczema or hand eczema [1]. Although there is no mortality directly associated with AD, this condition substantially impacts patients’ quality of life. In addition to the social stigmatization due to visible skin lesions, severe pruritus can disrupt sleep in patients with AD, which can lead to psychosocial comorbidities, including depression, anxiety, and suicidal ideations [2]. The economic burden of AD is attributed to not only direct medical costs but also to costs through lost work productivity. For instance, in Europe, the direct annual costs have been estimated to €7,000, while the indirect costs ranged from €7,000 to €14,000 per patient with severe AD [2].” The two references have been added into the References section.
Comment #2: In the section where the authors talk about possible tereputic targets, I suggest to the authors to consult and cite two works: "Fabbrocini G, Napolitano M, Megna M, Balato N, Patruno C. Treatment of Atopic Dermatitis with Biologic Drugs. Dermatol Ther (Heidelb ). 2018 Dec; 8 (4): 527-538. Doi: 10.1007 / s13555-018-0258-x. Epub 2018 Sep 4. PMID: 30182182; PMCID: PMC6261117. ", And" Napolitano M, Marasca C, Fabbrocini G, Patruno C. Adult atopic dermatitis: new and emerging therapies. Expert Rev Clin Pharmacol. 2018 Sep; 11 (9): 867-878. Doi: 10.1080 / 17512433.2018.1507734. Epub 2018 Aug 9. PMID: 30073901. "
Answer to comment #2
Following Reviewer’s suggestion, we have cited the works of two authors on lines 186-191 as follows: “Moreover, baricitinib, a selective JAK1 and JAK2 inhibitor reduces pruritus and inflammation in patients with moderate-to severe AD [48]. In addition, IL-13 antibodies, lebrikizumab and tralokinumab, improve AD symptoms although they do not show strong effect on pruritus [49,50]. Overall, more biologic drugs blocking IL-4 and IL-13 are under development and will be available in the future for AD treatment [51-53].”
Reviewer 2 Report
The article is an interesting narrative review on the development of itch in atopic dermatitis and on its possible treatments.
Although the article is complete, I think a revision is needed due to the lack of any material and methods section that explains how studies were screened and selected to be inserted in this review.
Also some minor queries:
Page 5 line 167; at the end of the paragraph add: "Other drugs blocking all these cytokines are under development and will be available in the future for AD treatment" and cite: "Dattola A, Bennardo L, Silvestri M, Nisticò SP. What's new in the treatment of atopic dermatitis? Dermatol Ther. 2019 Mar;32(2):e12787."
Page 1 line 35 please add: "some clinical phenotypes of AD are characterized by severe itch" and cite "Nettis E, Ortoncelli M, Pellacani G et al. A Multicenter Study on the Prevalence of Clinical Patterns and Clinical Phenotypes in Adult Atopic Dermatitis. J Investig Allergol Clin Immunol. 2020;30(6):448-450."
Thank You
Author Response
Comments and Suggestions for Authors
The article is an interesting narrative review on the development ofitch in atopic dermatitis and on its possible treatments.
Answer
We would like to thank Reviewer 2 for taking his/her time to read our manuscript, and for kind appreciation on this work. Blow, we provide point-by-point responses to your comments.
Major comment #1:
Although the article is complete, I think a revision is needed due to the lack of any material and methods section that explains how studies were screened and selected to be inserted in this review.
Answer to major comment #1
We agree with Reviewer’s suggestion. In this revised version, we have added Methods section on lines 58-65 as follows “For the current review, relevant literature published in English was searched using PubMed, Google Scholar and Web of Science to identify systematic articles published prior to January 2021. The terms used for retrieval are “dry skin”, “intractable itch”, “itch”, “pruritus”, “keratinocyte”, “sensory nerve fiber” etc. combined with “atopic dermatitis” as keywords. We screened the articles to select those published in international peer reviewed journals or books. Discrepancies in the assessment were resolved through discussion among the authors.”
Minor comment #1:
Page 5 line 167; at the end of the paragraph add: "Other drugs blocking all these cytokines are under development and will be available in the future for AD treatment" and cite: "Dattola A, Bennardo L, Silvestri M, Nisticò SP. What's new in the treatment of atopic dermatitis? Dermatol Ther. 2019 Mar;32(2):e12787."
Answer to minor comment #1
Following Reviewer’s suggestion, we have cited the above reference on lines 186-191 that “Moreover, baricitinib, a selective JAK1 and JAK2 inhibitor reduces pruritus and inflammation in patients with moderate-to severe AD [48]. In addition, IL-13 antibodies, lebrikizumab and tralokinumab, improve AD symptoms although they do not show strong effect on pruritus [49,50]. Overall, more biologic drugs blocking IL-4 and IL-13 are under development and will be available in the future for AD treatment [51-53].”
Minor comment #2:
Page 1 line 35 please add: "some clinical phenotypes of AD are characterized by severe itch" and cite "Nettis E, Ortoncelli M, Pellacani G et al. A Multicenter Study on the Prevalence of Clinical Patterns and Clinical Phenotypes in Adult Atopic Dermatitis. J Investig Allergol Clin Immunol. 2020;30(6):448-450."
Answer to minor comment #2
Thank you for your suggestion. In this revised version, we have added Nettis work on lines 36-37 as “The most frequent clinical phenotype of AD is lichenified/exudative flexural dermatitis alone or associated with head/neck eczema or hand eczema [1].
Round 2
Reviewer 2 Report
The authors responded to all queries. The article is ready to be published.
Author Response
Thank you very much for the reviewer's comment